# A Quasi-Intelligent Maritime Route Extraction from AIS Data

**DOI:** 10.3390/s22228639

**Published:** 2022-11-09

**Authors:** Shem Otoi Onyango, Solomon Amoah Owiredu, Kwang-Il Kim, Sang-Lok Yoo

**Affiliations:** 1College of Ocean Sciences, Jeju National University, Jeju 63243, Korea; 2Department of Marine Engineering and Maritime Operations, Jomo Kenyatta University of Agriculture and Technology, Nairobi P.O Box 62000-00200, Kenya; 3Future Ocean Information Technology, Jeju 63208, Korea

**Keywords:** automatic identification system (AIS), clustering, route planning, waypoint discovery, traffic network, algorithm

## Abstract

The rapid development and adoption of automatic identification systems as surveillance tools have resulted in the widespread application of data analysis technology in maritime surveillance and route planning. Traditional, manual, experience-based route planning has been widely used owing to its simplicity. However, the method is heavily dependent on officer experience and is time-consuming. This study aims to extract shipping routes using unsupervised machine-learning algorithms. The proposed three-step approach: maneuvering point detection, waypoint discovery, and traffic network construction was used to construct a maritime traffic network from historical AIS data, which quantitatively reflects ship characteristics by ship length and ship type, and can be used for route planning. When the constructed maritime traffic network was compared to the macroscopic ship traffic flow, the Symmetrized Segment-Path Distance (SSPD) metric returned lower values, indicating that the constructed traffic network closely resembles the routes ships transit. The result indicates that the proposed approach is effective in extracting a route from the maritime traffic network.

## 1. Introduction

Rapid economic growth and increased international trade have increased maritime transport, resulting in complex maritime traffic networks in restricted waters. With the rapid development of water transportation, the navigation environment is becoming increasingly complex, as is the risk of water traffic incidents, which thus increases both the requirement and difficulty for traffic controllers and ship operators. As a result, it is critical to scientifically understand the transportation system in marine areas in order to further ensure navigation safety [1]. The mandatory installation of an Automatic Identification System (AIS), an automatic tracking transponder that in specified time intervals sends information about the ship’s static and dynamic identification, generates a large amount of ship trajectory data globally. These data provide latitude, longitude, speed, and course information for maritime pattern extraction and vessel behavior prediction. However, the determination of an approach to deeply examine these AIS data which allows the identification of a ship’s behavior pattern presents a critical task. As established by the International Maritime Organization’s Convention for the Safety of Life at Sea (SOLAS), an AIS transponder must be on board all ships with a gross tonnage of 300 or more that sail in international waters, ships with a gross tonnage of 500 tons or more that do not sail in international waters, and passenger ships of any size [2]. Terrestrial and satellite networks of AIS receivers provide an ever-increasing volume of maritime traffic data, which can enhance general awareness by building patterns of vessel activities in both coastal and open waters [3]. AIS, as the primary source of ship data, has often been used for ship collision modeling and navigation risk assessment and management, to enhance navigation safety [4]. In addition, the availability of AIS data has sparked interest in employing Artificial Intelligence (AI) techniques in ship route planning and analysis of complex marine traffic networks, thereby improving navigation efficiency and safety.

Planning a safe and optimal route for a ship presents an essential and challenging task [5]. A safe and efficient vessel operation requires prescient berth to berth route planning which is always conducted manually by a navigation officer with the help of existing bridge support systems and foraging an enormous amount of navigation-related materials, such as sailing directions, a guide to port entry, notice to mariners, and many others. Even for an experienced mariner, it is a time- and energy-consuming process [6,7]. Furthermore, a navigation officer unfamiliar with a sea area is unlikely to have information on previous experiences or best practices in the waters under consideration. This issue can be resolved by implementing a system that provides a network of traffic routes based on the past behaviors of similar ships that have traveled in the considered waters. Such a network can be constructed automatically from past trajectories of ships’ AIS data.

AI techniques such as Genetic Algorithm (GA) [8,9], Particle Swarm Optimisation (PSO) [9], Ant Colony Algorithm (ACA) [10], clustering algorithms [3,7,11,12], and Artificial Neural Network (ANN) [1] are currently gaining popularity due to their potential in extracting maritime routes from AIS big data. Several researchers have conducted empirical studies on naval routing and route planning. For example, Filipiak et al. [8] used the Cumulative Sum (CUSUM) algorithm to extract maneuvering points, the GA algorithm to discover waypoints, and the graph algorithm to detect edges between waypoints. The graph nodes represent waypoints and connected edges represent maritime roads. From density estimation, Lee et al. [11] extracted the polygon structure of the main routes commonly used in Korean waters. The extracted routes were divided into three categories: main route, branch route, and inner branch route. Zhang et al. [7] used the Douglas–Peucker (DP) algorithm to simplify ship trajectories by identifying characteristic points clustered by the Density-Based Spatial Clustering of Applications with Noise (DBSCAN) algorithm to extract turning nodes. Following that, the ACA algorithm was used to determine the best route from the turning nodes. Forti et al. [3] generated a maritime traffic pattern via a compact graph-based model. Using an Ornstein–Uhlenbeck mean-reverting stochastic process (OU), researchers identified navigational change points, clustered the change points using the DBSCAN algorithm to identify waypoints, and constructed marine traffic patterns from the waypoints using a directed weighted network. Yan et al. [12] transformed ship trajectory data into a ship trip semantic object (STSO) with semantic information, each ship trip abstracted as a stop-waypoint-stop trip object. In their research, the ordering points to identify the clustering structure (OPTICS) algorithm, clusters the stop and moving points to form waypoints and maritime routes extracted from graph theory. Finally, Wen et al. [1] employed a supervised machine-learning algorithm, ANN, to extract maritime routes. First, Wen and colleagues clustered turning areas to extract the turning region using the DBSCAN algorithm. After that, the ANN was trained to learn the relationship between the turning regions and generate a reasonable route. A summary of studies applied by various authors using AI techniques in route extraction is summarized in Table 1. Most of the previous studies focus on the visualization of traffic routes which is beneficial in the management of maritime traffic for the given area. Despite their ability to properly reflect and portray actual traffic, they cannot be relied upon for route planning as the quantitative aspect and classification by ship characteristics of the traffic network are not presented.

The goal of this study’s proposed approach is to generate a traffic network representing maritime routes quantitatively, by ship characteristics such as length, and type. The resulting maritime route should reflect the accurate depiction of maritime routes in the considered waters, such as local Traffic Separation Schemes (TSS). The generated maritime traffic routes are calculated from one-year AIS data. The technique comprises three components: a course variance algorithm as a pre-processing step for identifying maneuvering points, a Hierarchical Density-Based Spatial Clustering of Applications with Noise (HDBSCAN) clustering of extracted maneuver points for waypoint discovery, and a network algorithm for recognizing edges between waypoints. The network algorithm assigns a weighted value to the edges according to selected ship characteristics. Thus, the result can be used to plan a maritime route effectively.

## 2. Materials and Methods

Understanding ship movement patterns and operational elements are required to construct a maritime traffic route. Ships sail routes as recommended by navigation charts, marine pilots, and vessel traffic service (VTS) in order to reduce trip distance and ensure navigation safety. As a result, most merchant ships have similar trajectories in navigable waters, which can be analyzed to generate maritime routes. Figure 1 depicts the architecture and implementation of the proposed maritime route construction method.

The architecture is divided into four essential parts: AIS data mining, which involves pre-processing trajectories based on the ship’s unique identifier, Maritime Mobile Service Identifier (MMSI) number; course variance analysis, which identifies maneuvering points; waypoint discovery via the HDBSCAN algorithm; and maritime traffic network construction. Subsections in Section 2 contain detailed information on the proposed approach.

### 2.1. AIS Data Cleaning

Ship maneuvering is a complex but slow dynamic operation, and there may be lengthy delays between the initial commencement of a maneuver command and the ship’s motion. A suitable high-frequency of maritime observations is required for the detection of the maneuver’s first conscious motion [13]. The AIS transmits data every 2 to 180 s, depending on the ship’s dynamic state, as shown in Table 2.

The trajectory of one ship *j* in its raw form can be described as a finite sequence of observations denoted as $T_{j}=\left\{\left(x_{j}^{1},t_{j}^{1}\right),\left(x_{j}^{2},t_{j}^{2}\right),...,\left(x_{j}^{m},t_{j}^{m}\right)\right\}$, where $x_{j}^{i}$ is defined by Equation (Equation 1) and $t_{j}^{i}$ is the timestamp [14]. In this study, we modify the trajectory in its raw form to adapt it to our study approach [15].
(1)$$\begin{array}{c} T_{j}^{i}=\left\{MMSI_{j},\theta_{j}^{i},Lat_{j}^{i},Lon_{j}^{i}\right\};\;\;\;i=1,2,...,m \end{array}$$
where *i* is the sequence index number of a ship trajectory, *m* is the total length of a ship trajectory, $\theta_{j}^{i}$ is ship course over ground (COG), $Lat_{j}^{i}$ and $Lon_{j}^{i}$ is latitude, and longitude coordinate at the time $t_{j}^{i}$.

AIS data cleaning entails pre-processing the trajectory data extracted from the shore database. The first step is to combine the ship’s dynamic data with static data, utilizing the ship’s MMSI as the unique identification, and then sorting the AIS messages in ascending timestamp order. The ship’s unique information is represented by static data, which comprises the MMSI, name, type, International Maritime Organization (IMO) number, callsign, length, and draft. Dynamic data represents data that vary while a ship operates, and includes the time and date, latitude, longitude, speed, COG, and direction. The second step entails deleting duplicated trajectories, trimming anomalous trajectories with a time threshold, and reorganizing the trajectories by time. Structured Query Language (SQL) was utilized to extract AIS from the shore database, and Python was used to clean the data.

### 2.2. Course Variance Analysis

Ships alter course in accordance with the International Regulations for Preventing Collisions at Sea 1972 (COLREG) under the following conditions: Rule 8b, to avoid a collision; Rule 9a, to maintain the recommended channel route; and Rule 10b, when joining or leaving a traffic separation scheme [16]. Individual ships’ sequential actions to change course in line with Rules 9a and 10b produce trajectories that provide critical information about traffic behavior and the route network in considered waterways.

The input to the quasi-intelligent maritime route extraction model is a time-series trajectory of critical ship positions. The course difference between two points in the AIS trajectory can be calculated using Equation (Equation 2) [17].
(2)$$\begin{array}{c} \Delta \theta_{j}^{i}=\{\begin{array}{ccc} 360-|\theta_{j}^{i+1}-\theta_{j}^{i}| & \; & if\;|\theta_{j}^{i+1}-\theta_{j}^{i}|\geq 180^{\circ }\\ |\theta_{j}^{i+1}-\theta_{j}^{i}| & \; & otherwise \end{array} \end{array}$$

The course difference of a trajectory can be expressed by the set $\{\Delta \theta_{j}^{i}{\}}_{i=1}^{m-1}$.

Figure 2 demonstrates the extraction of turning points from a ship’s successive course alteration. The green circle represents turning points. Given $\theta_{j}^{i}$ is the initial course of *i*-th turning point, taking the direction of the turning point at $t_{j}^{i}$, the $\theta_{j}^{i+1}$ is the end course of the *i*-th turning point and the succeeding starting course at $t_{j}^{i+1}$ in which course alteration ends at course $\theta_{j}^{i+2}$. In order to obtain the macroscopic changing of course alteration curve and remove local variation and noise which may be caused by equipment error, environmental influence, autopilot, etc., $\Delta \theta_{j}^{i}$ should be smoothed before the next step. The smoothing factor is the simple moving average as shown in Equation (Equation 3). Given a sequence $\{\Delta \theta_{j}^{i}{\}}_{i=1}^{m-1}$, an *n*-moving average is a new sequence $\{\Delta \theta_{j,n}^{k}{\}}_{i=1}^{m-n+1}$ defined from the $\Delta \theta_{j}^{i}$ by taking the arithmetic mean of subsequence of *n*-terms.
(3)$$\begin{array}{c} \Delta \theta_{j,n}^{k}=\frac{1}{k}\sum_{k=i}^{i+n-1}\Delta \theta_{j}^{k} \end{array}$$
where *n* is the window size, *k* is the sequence index number of trajectory observations. The smoothed set $\Delta \theta_{j,n}^{k}$ is divided by a set of sequential finite observations from the trajectory in seconds to give the rate of turn ζ which represents the threshold. The model’s sensitivity is controlled by the threshold ζ. The lowest number of ζ denotes higher sensitivity, while the highest value represents reduced sensitivity, with the possibility of missing important maneuvers. The model detects more maneuver points as it becomes increasingly sensitive.

The above steps are formalized in a course variance algorithm, which is then executed in Python to produce a sample, Figure 3, that shows an example of maneuver points extracted from an AIS trajectory.

### 2.3. HDBSCAN Clustering of Maneuver Points

Cluster analysis is an unsupervised machine-learning task of arranging a set of similar data points into groups to explore and analyze patterns of similarity in a dataset. A good clustering method should be stable in order to limit results disparity caused by changes in a few samples, as well as minimize manual intervention to maintain objectivity of the results. DBSCAN is a popular clustering paradigm, but its major weakness is that its ε (epsilon) parameter serves as a global density threshold and it is therefore not possible to discover clusters of variable densities or nested clusters beyond ε [18]. HDBSCAN was proposed as an improved extension of DBSCAN for data exploration across diverse research fields and different density thresholds [19]. The algorithm requires a *min cluster size* as user input and then simplifies a complex single-linkage hierarchy to a smaller tree of candidate clusters [18]. A flat solution is extracted automatically based on local cuts as illustrated in Figure 4b.

The following are key descriptors in the HDBSCAN algorithm. *Core distance*: the distance between the sample point and the $g_{th}$ nearest sample point as shown in Figure 4a. *Mutual reachability distance*: the maximum value of core distance *a*, core distance *b*, or the distance between *a* and *b* for two points *a* and *b* as illustrated in Equation (Equation 4) and Figure 4a.
(4)$$\begin{array}{c} d_{reach-g}\left(a,b\right)=max\{core_{g}\left(a\right),core_{g}\left(b\right),dist\left(a,b\right)\} \end{array}$$
where dist(a,b) is Euclidean distance between point *a* and point *b*. The hierarchy is constructed based on the *mutual reachability distance* between points. Python, a well-known data mining and machine learning language, has a well-established HDBSCAN algorithm sci-kit implementation [20]. The python HDBSCAN algorithm takes two inputs: *min cluster size* and *min samples* [19]. The former is the core parameter of HDBSCAN and represents the minimum size of clusters. The larger the parameter, the less clustered species there will be, and fewer points than this will be deemed ’noise’. The latter is defined as the number of samples in a neighborhood for a point to be regarded as a core point. The core parameter, *min cluster size*, is approximated using clustering performance metrics in Section 2.4. The maneuver points are grouped into clusters that have a measure of central tendency, the centroid. Equation (Equation 5) is used to calculate the centroids [X,Y] from HDBSCAN clusters, which serve as the waypoints.
(5)$$\begin{array}{c} X,Y=\sum_{c=1}^{r}\frac{x_{c},y_{c}}{r};\;\;\left(x_{1},y_{1}\right),\left(x_{2},y_{2}\right),...,\left(x_{r},y_{r}\right) \end{array}$$
where *x* and *y* are the latitude and longitude of a maneuver point in a cluster, and *r* is the total number of maneuver points in a cluster.

### 2.4. Clustering Performance Metrics

There are approaches for estimating *min cluster size* from a dataset, such as the popular elbow method, which is fairly subjective and thus unreliable when employed solely. Taking into account the objectivity of clustering performance metrics, the Silhouette Coefficient (SC) [21] and the Davies–Bouldin index (DBI) [22] are used to evaluate the selection of the minimum cluster size. SC measures the tightness and separation of points in the same class compared with points in different classes. The SC value is evaluated as shown in Equation (Equation 6). An SC score of 1 indicates that the sample is distinct from the adjacent class and clearly distinguished, indicating that it has a good clustering effect, whereas a score of 0 indicates that the sample is on the decision boundary of two adjacent classes, and a negative value indicates that the sample is classified incorrectly.
(6)$$\begin{array}{c} SC=\frac{1}{P}\sum_{p=1}^{P}\frac{d(p)-c(p)}{max\{d(p),c(p)\}};\;\;-1\leq SC\leq 1 \end{array}$$
where c(p) is the average distance between *p* and all other points in its own cluster, and d(p) is the average distance between *p* and all other clusters except its own cluster. To calculate DBI, the sum of the average intra-cluster dispersion between samples of two clusters, $S_{p}+S_{q}$, is divided by the distance between the center points of two clusters $M_{pq}$ as indicated in Equation (Equation 7).
(7)$$\begin{array}{c} DBI=\frac{1}{P}\sum_{p=1}^{P}max\{\frac{S_{p}+S_{q}}{M_{pq}}\};\;\;p\neq q \end{array}$$

The smaller the DBI value, the better the clustering effect. A comprehensive clustering performance meter (CCPM) is developed to evaluate clustering results based on SC and DBI scores as defined by Equation (Equation 8) [23].
(8)$$\begin{array}{c} CCPM=SC+\frac{1}{DBI} \end{array}$$

The larger the CCPM value, the better the clustering effect.

### 2.5. Maritime Traffic Network

The maritime traffic network is constructed based on graph theory. A graph G=(V,E) consists of a set of objects *V* called vertices and other sets *E* whose elements are called edges [24]. The centroids of clusters from the HDBSCAN algorithm generate a set of waypoints which are the vertices of the maritime traffic network. Therefore, we need a method to discover the edges that connect the waypoints. The historical AIS data is separated according to an individual MMSI. For each MMSI, the trajectory observations are assigned a waypoint label in accordance with the ’visited’ cluster label. This process of adding information, waypoint label, and distance to the label, into the trajectory observation is referred to as trajectory enrichment.

Having assigned the waypoints to the trajectory observations, the next step is the extraction of a ship’s route by waypoint labels, which are stored as dictionaries. This process involves ordering a ship’s trajectory enriched observations by time, $t_{j}^{i}$, thereby sequencing the waypoints in chronological order of the ship’s movement. The dictionary’s keys represent a ship’s MMSI, while the dictionary’s values are a list of sequential waypoints.

The generated dictionary is employed to create a data frame that is used to extract the network edges in accordance to ship characteristics. The network edge is presented as a line segment characterized by its starting point and ending point together with its associated features such as latitude, longitude, and waypoint label. The weighted feature of an edge is computed from the cumulative count of ships by length, type, or both. The segments are assigned a color according to the weighted value and each is plotted to form a maritime traffic network depicting quantitative traffic flow in the target waters. The maritime traffic network construction procedure is detailed as Appendix A.

The quality of the maritime traffic network is dependent on the quality of AIS data. Short trajectories are eliminated when creating the route dictionary by setting the value list threshold to above five sequential waypoints. Edges that are ’unpopular’, that is, bearing a low cumulative value, are filtered out in the visualization of the maritime traffic network.

## 3. Case Study

In this section, a case study on the maritime traffic network is illustrated to verify the proposed approach. The quality of results depends on the selection of parameters, and the quality of historical AIS data. Ships rarely perform maneuvers in deep waters due to the availability of spatial freedom. Since our study is anchored on maneuver point detection and clustering, the port approach provides a suitable target area for the study. Therefore, the Incheon port approach in the Republic of Korea was selected as the target area for the study, bearing the following geo-location boundary information, latitude: $36.5^{\circ }$ N to $37.7^{\circ }$ N, longitude: $125.5^{\circ }$ E to $127.1^{\circ }$ E. In the Incheon Port approach, the navigation channels are created based on the concept of harbor limit. This is divided into the inner harbor limit and outer harbor limit, while the point where the navigation channels meet represents a sea area that requires attention due to the complicated maritime traffic involving various traffic flows [25].

Figure 5 shows that the Incheon port is accessed through the inbound route, Dong-sudo. Outbound ships leave the port via the Seo-sudo route into the open waters where inbound traffic from Heuk-do TSS and outbound traffic from Pyeongtaek port converge. The Incheon port approach area harbors three ports: Incheon port, Pyeongtaek port, and Daesan port, each of which has its own dedicated inbound–outbound fairway.

In this case study, one year of historical AIS data collected from the Incheon port approach area from 2018.01.01 to 2018.12.31 is used, as summarized in Table 3. We examined AIS data from ocean-going ships and applied a speed threshold to exclude vessels in berth and anchorage. The AIS data are pre-processed and used as input into the course variance analysis algorithm to produce maneuver points. There is no scientific method of selecting the algorithm sensitivity threshold ζ to give an optimal plot of maneuver points. Therefore, an empirical value is chosen in accordance with the visualized maneuver points scatter plot. The lowest sensitivity threshold value of ζ=0.05 was chosen for this case study from a possible range of 0.05≤ζ≤4.78 computed from the course variance algorithm.

### 3.1. HDBSCAN Clustering

A mesh grid density plot of the maneuver points in Figure 6 shows a varied density distribution of the maneuver points in the Incheon approach area. The mesh grid color bar value of 1 indicates a densely populated grid, and 0 indicates a sparsely populated grid. The varied distribution of the maneuver points in the port approach area is an indication that an attempt to cluster the maneuver points using a global threshold, ε, from DBSCAN will return a small number of clusters for a lower ε value, thereby excluding more maneuver points as noise, and huge clusters for a higher ε value, losing the traffic flow behavior and information in the target area.

The HDBSCAN algorithm is suitable for discovering varied density clusters in the Incheon port approach area. Experience and subjective judgment can be relied upon in determining the core input, *min cluster size*, to the HDBSCAN algorithm. However, the aforementioned methods lead to a loss of generality and objectivity in the information from the clusters. As explained in Section 2.4, a clustering performance metric is used to obtain a close estimate of the *min cluster size*.

A range from 0 to 400 is optimized in the three clustering measurement criteria, SC,DBI, and CCPM, to select the optimal *min cluster size* as the HDBSCAN algorithm’s input value. The optimal value of *min cluster size* is the highest CCPM score at a stable SC state and the lowest DBI index. It can be observed from Figure 7a that the SC is relatively stable in the range of 90 to 150 cluster sizes, with the minimum DBI value at 119 with a score of 1.15.

The correlation between the number of clusters and cluster size in Figure 7b gives an ’elbow’ curve that is deducible by the elbow method. As the number of clusters increases, the cluster size decreases sharply at first, then turns into a turning region, and thereafter stabilizes in the region of 100 to 190 cluster size. The curve continues downward and flattens as the cluster size increases. As a result, the stabilized region of the SC score in Figure 7a is correspondingly within the stabilized region of Figure 7b. Therefore, the elbow method is used to validate the choice of optimal value *min cluster size* from the cluster performance metrics method.

An optimal value of 119 is selected for the *min cluster size* as it is recommended to choose the smallest value of the core parameter of HDBSCAN in order to cluster the majority of the points. Python implementation of the HDBSCAN algorithm requires two parameters as inputs: *min cluster size* and *min samples*. The *min samples* provide a measure of clustering conservation. The larger the value of *min samples*, the more conservative the clustering, the more points that will be declared as noise, and clusters will be restricted to progressively denser areas [26]. Therefore, a smaller intuitive value of *min samples* = 1, which returns the best clustering effect, is assigned for illustration purposes so as to minimize the number of maneuver points considered as noise. The maneuver points in Figure 7c are clustered by HDBSCAN to give 74 clusters, excluding noise, as shown in Figure 7d. The waypoint of each of the 74 clusters is computed from the mean of maneuver points within the same cluster and labeled as per the cluster label.

### 3.2. Maritime Traffic Network

The quality of the maritime traffic network depends on the quality of the discovered waypoints from the HDBSCAN algorithm and the quality of edges from the network algorithm. AIS trajectories with large time differences between observations due to signal loss, or sliced data from the database, can result in the discovery of poor quality edges and impossible edge connections. To avoid this phenomenon, a time threshold is used to pre-process the historical AIS data so as to extract trajectories with continuous observations. Each ship’s trajectory data are separated into subcategories with a different MMSI identifier. If the set time threshold within the observations is exceeded, the first part of the subcategory continues to use the original MMSI while the subsequent subcategories are assigned a new MMSI number ’431456000XXX’, assuming the original MMSI number, for example, is ’431456000’. The pre-processed AIS data are processed in the network algorithm to construct a maritime traffic network as per ship type in Figure 8 and ship length in Figure 9.

The color bars in the sub-figures represent ship traffic by MMSI count, with red representing the highest value and white representing the lowest value in the illustrated category. Figure 8a shows a maritime traffic network for all ships. It is observed that the highest ship traffic is in Heuk-do TSS at above 4000, which is the outbound traffic, with a majority in the region of above 3000 destined to Incheon port from inbound Heuk-do TSS via Dong-sudo to Incheon port, 1500 to 1000 in Pyeongtaek port, and a minority of below 500 in Daesan port. Passenger ship traffic in Figure 8b operates to and from the Incheon passenger terminal, Pyeongtaek passenger terminal, and Deokjeok Island. The highest passenger ship traffic is observed at Seo-sudo and Dong-sudo lanes in the Incheon port, with the lowest traffic from Deokjeok Island. Passenger ships outbound and inbound traffic are observed from the west of the Incheon port approach area. In Figure 8c, tanker ship traffic is highest at Heuk-do TSS outbound lane, above 1400, with a large portion of tanker traffic observed to and from Incheon port and a minority of traffic, below 600, at Pyeongtaek and Daesan port. In Figure 8d, cargo ship traffic is highest between Heuk-do TSS and Incheon port, with Pyeongtaek port ranking second and being the only other port with cargo ship traffic. Most cargo inbound and outbound traffic in the Incheon port approach is at Heuk-do TSS.

Ship traffic by ship length in Figure 8a–c shows that traffic is concentrated in the Incheon port’s approach Heuk-do TSS, Seo-sudo and Dong-sudo lanes, with the majority of ships in the 100–200 m LOA category.

## 4. Discussion

Since routes are generally computed based on fuel consumption and navigational safety, the majority of ship mobility is quite regular. These preferred routes experience a greater volume of traffic than others. The construction of a maritime traffic network from traffic behavior using AIS data is crucial for gaining an understanding of ship traffic patterns for the design and risk assessment of safe routes. In order to verify the effectiveness of the proposed approach, the maritime traffic network is compared with the ground truth AIS trajectories, Figure 10. The AIS grid plot is interpreted from the color bar as high-traffic areas having a grid color of red and a value of 1, while low-traffic areas have a grid color of white and a value of 0.

The Symmetrized Segment-Path Distance (*SSPD*) proposed by Besse [27] provides the best criteria for verification. The definition of *SSPD* is based on the Point-to-Segment distance, $D_{pt}$, from a point *p* to a trajectory *T*. It represents the minimum distances between this point and all segments *s* that compose *T*. The Segment-Path distance from trajectory $T^{1}$ to trajectory $T^{2}$ is the mean of all distances from points composing $T^{1}$ (traffic edges) to the trajectory $T^{2}$ (AIS trajectories) as demonstrated in Equation (Equation 9). *SSPD* metric dictates that the smaller the $D_{SSPD}$ values, the better the comparison.
(9)$$\begin{array}{cc} \; & D_{SPD}\left(T^{1},T^{2}\right)=\frac{1}{n_{1}}\sum_{i_{1}=1}^{n_{1}}D_{pt}\left(p_{i_{1}}^{1},T^{2}\right),\\ \; & \mathrm{where},D_{pt}\left(p_{i_{1}}^{1},T^{2}\right)=min_{i_{2}\in \left[0,...,n_{2}-1\right]}D_{ps}\left(p_{i_{1}}^{1},s_{i_{2}}^{2}\right),\\ \; & D_{ps}=\left\{\begin{array}{cc} hav\left(p_{i_{1}}^{1},p_{i_{1}}^{1proj}\right) & if\;p_{i_{1}}^{1proj}\in s_{i_{2}}^{2},\\ min\{hav\left(p_{i_{1}}^{1},p_{i_{2}}^{2}\right),hav\left(p_{i_{1}}^{1},p_{i_{2}+1}^{2}\right)\} & otherwise, \end{array}\right.\\ \; & D_{SSPD}\left(T^{1},T^{2}\right)=\frac{D_{SPD}\left(T^{1},T^{2}\right)+D_{SPD}\left(T^{2},T^{1}\right)}{2} \end{array}$$
where $p_{i_{1}}^{1proj}$ is the orthogonal projection of $p_{i_{1}}^{1}$ in $s_{i_{2}}^{2}$, and hav denotes the Haversine distance between two points on a sphere. Four prominent routes within the Incheon port approach area are sampled for evaluation: Heuk-do TSS to Incheon port (HTSS-INCH), Incheon port to Heuk-do TSS (INCH-HTSS), Heuk-do TSS to Daesan port (HTSS-DSN) and Pyeongtaek port to South West of port approach area (PYTK-SWST).

From Table 4, we can conclude that routes from the proposed maritime traffic network are reliable in route planning, as evidenced by smaller *SSPD* values in comparison with the AIS trajectories. Segments 6–7–5 of the PYTK-SWST route returned the largest *SSPD* value in the test due to the possibility of being in deep, unrestricted waters far from the coast. The segments 70–72–73–71–69–66–65–62–56–61–67–68 in Figure 10 are part of a two-way traffic Masan-sudo channel. The densely spaced waypoints are a result of waypoint identification of the course-keeping maneuvers from both inbound and outbound traffic at the channel bends. Masan-sudo: 2–3–4–41, Seo-sudo: 15–20–21–16–26, and Dong-sudo: 39–47–52–51–46–60–59, returned the smallest values of *SSPD*, affirming the proposition of the proposed approach in route extraction for voyage planning. The isolated waypoints 0, 1, 8, and 9, are minor local landing sites and routes with low traffic to construct a traffic network in relation to the area network, but significant maneuver points for waypoint identification.

In summary, compared with the existing methods, the proposed maritime traffic network for route extraction in this study bears the following distinct advantages. First, the proposed method provides a quantitative feature in addition to the visualization of routes in a traffic network, thereby making it possible to select prominent routes during route planning for safe navigation as compared to methods used by Yan et al. [12], Wen et al. [1] and Zhang et al. [7] which offer only visualization of the traffic network. Second, unlike the method proposed by Wen et al. [1], which focuses on route extraction by ship length, and the approach reported by Filipiak et al. [8], which constructs the traffic network by comparative sizes of ship width, the construction of the maritime traffic network by ship type, length, or both, make it convenient for extracting routes by the ship’s own characteristics. Third, the proposed method uses a single program, Python, to construct the traffic network, thereby simplifying further development in the future, as recommended by Lee et al. [25]. Therefore, this method has more distinctive advantages over existing methods, and is hence more effective in maritime traffic network construction for route planning.

## 5. Conclusions

A novel method based on maneuver points clustering and graph theory is proposed to construct a maritime traffic network from historical AIS data. A case study in the Incheon port approach area together with a comparison between the maritime traffic network and historical AIS trajectories using Symmetrized Segment-Path Distance verified the effectiveness of the proposed method. The novel ideas presented in this proposed method are as follows: The application of clustering performance metrics in the selection of HDBSCAN core parameter, *min cluster size*; Construction of a maritime traffic network to quantitatively display prominent routes based on ship characteristics such as ship type, length, or a combination of type and length; Combination of HDBSCAN clustering and graph theory to construct a maritime traffic network.

Maneuvering points from the historical AIS trajectories are extracted by the course variance algorithm. The HDBSCAN algorithm clusters the maneuvering points, suitable for clustering data with varying density thresholds, to discover waypoints. The waypoints as nodes and edges from the network algorithm are combined from graph theory to construct a maritime traffic network, the edges of which are weighted according to the selected ship characteristic and quantitatively displayed as a color bar for the selection of optimal routes during route planning. A case study on the Incheon port approach area is presented, and the maritime traffic network is evaluated by comparison with the historical AIS using Besse [27] *SSPD* metric.

The proposed method is titled ’quasi-intelligent’ because the traffic network construction algorithm does not autonomously determine its input core parameters. Therefore, the algorithm is interrupted on two occasions: when selecting sensitivity threshold ζ in the course variance algorithm, and when choosing *min cluster size* and *min samples* for the HDBSCAN algorithm.

The proposed method is unsuitable for route extraction in ships that seldom perform maneuvers since the study is anchored on maneuver point extraction. In addition, the method is susceptible to erroneous or missing AIS messages, which can be solved by trajectory reconstruction or interpolation. The proposed solution does not absolve the navigator from the responsibilities of route planning and route safety validation. In the future, more advanced criteria must be added to this study, such as dynamic weather routing, time minimization, and fuel-consumption routing.

## Figures and Tables

**Figure 1 sensors-22-08639-f001:**
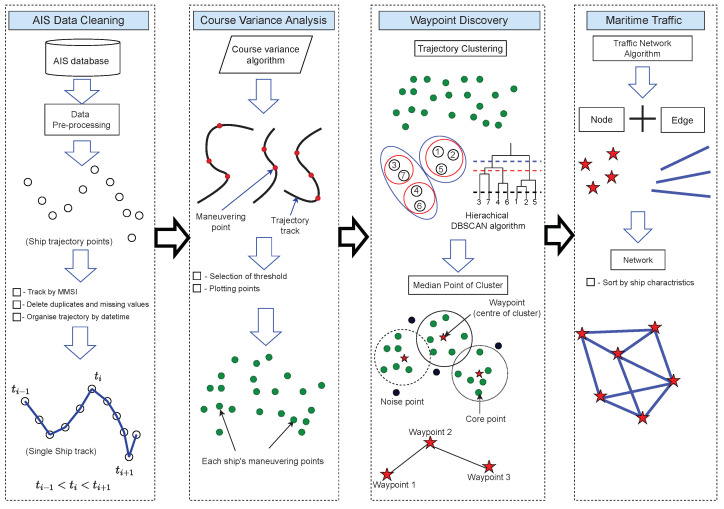
Maritime route extraction framework combining course variance analysis, clustering algorithm, and traffic network construction.

**Figure 2 sensors-22-08639-f002:**
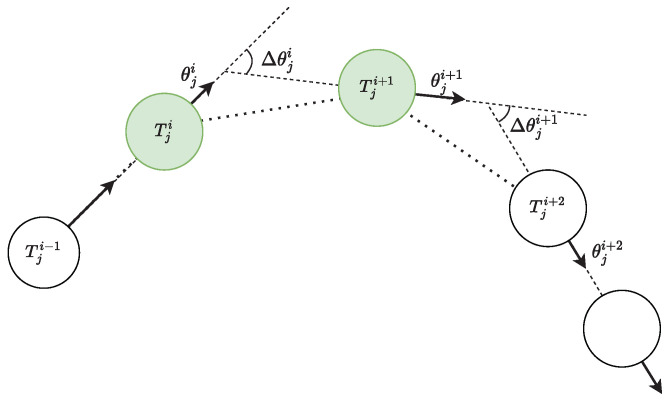
Turning points from course alteration.

**Figure 3 sensors-22-08639-f003:**
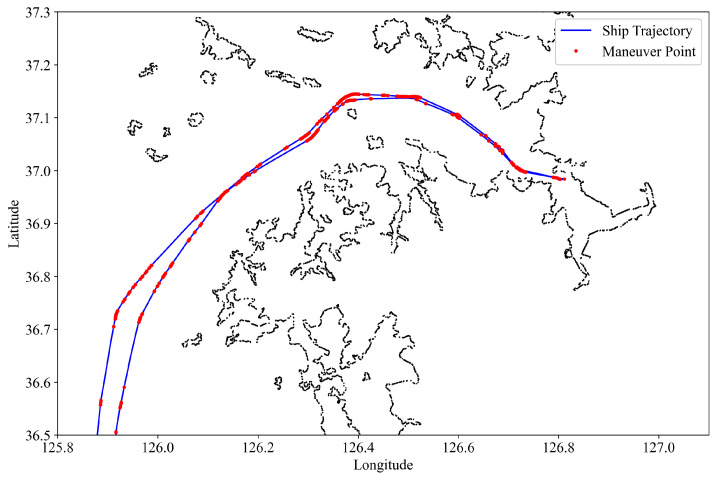
Maneuvers detected from course variance analysis.

**Figure 4 sensors-22-08639-f004:**
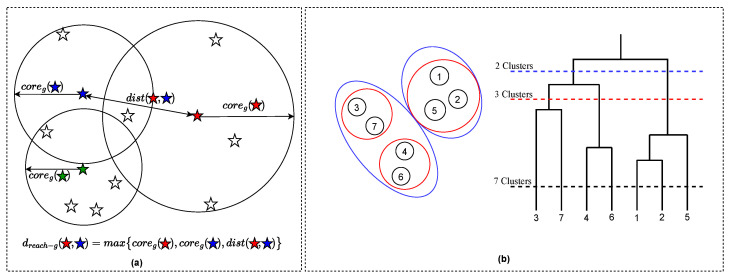
A concept of HDBSCAN clustering algorithm. (**a**) Mutual reachability distance computation from three clusters bearing varied densities. (**b**) Agglomerate dendrogram showing significant clusters that can be extracted.

**Figure 5 sensors-22-08639-f005:**
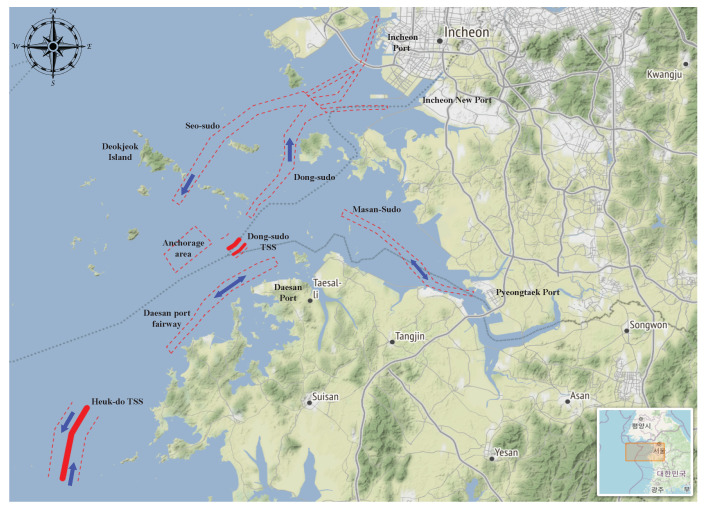
Overview of the Incheon port approach area.

**Figure 6 sensors-22-08639-f006:**
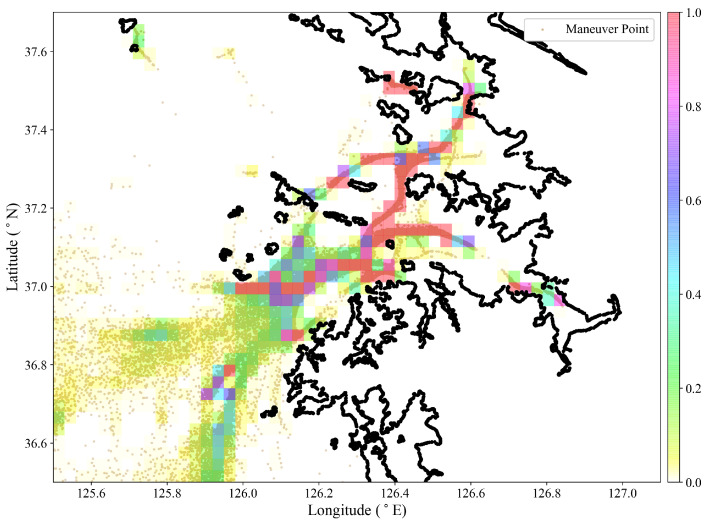
Mesh grid density plot superimposed over maneuver points scatter plot.

**Figure 7 sensors-22-08639-f007:**
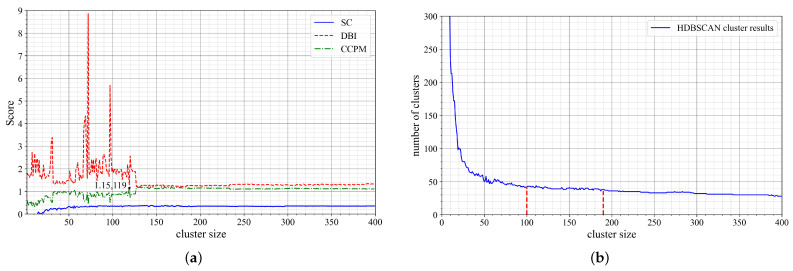
HDBSCAN clusters extracted after selection of optimal *min cluster size*. (**a**) Clustering performance metrics of HDBSCAN. (**b**) Elbow method; correlation between the number of clusters and cluster size. (**c**) Maneuver points scatter plot when ζ=0.05. (**d**) HDBSCAN clusters and waypoints.

**Figure 8 sensors-22-08639-f008:**
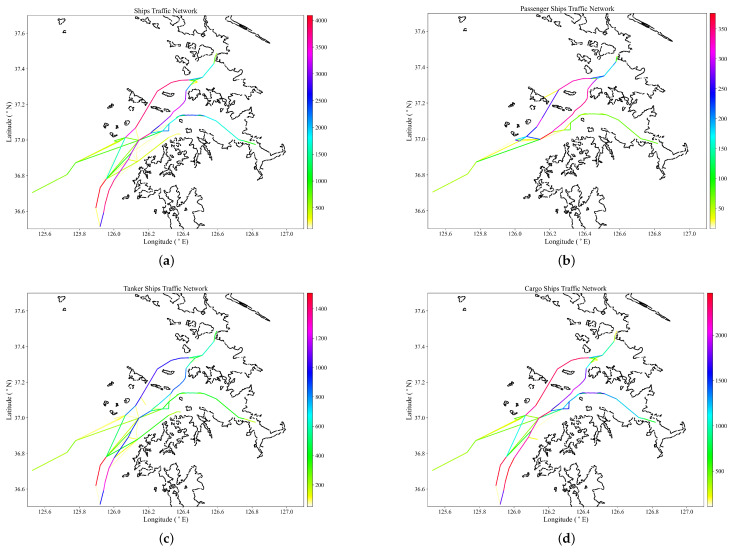
Maritime traffic network by ship type. (**a**) Traffic network for all ships. (**b**) Passenger ships traffic network. (**c**) Tanker ships traffic network. (**d**) Cargo ships traffic network.

**Figure 9 sensors-22-08639-f009:**
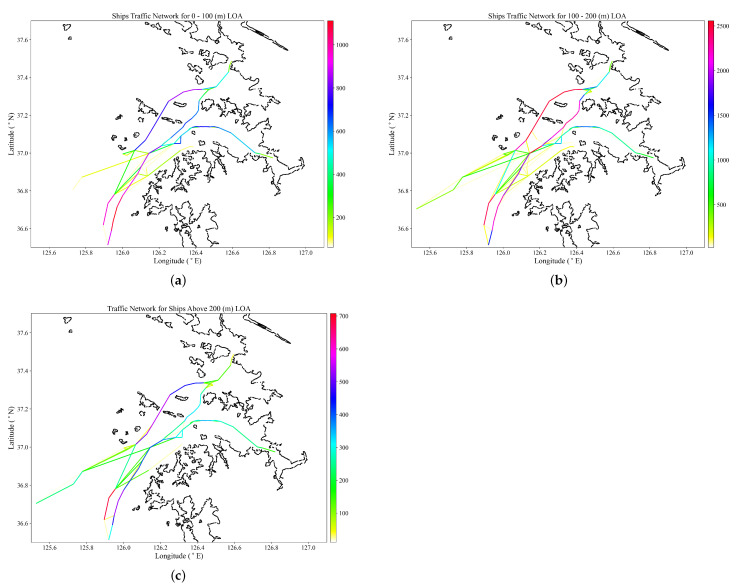
Maritime Traffic Network by ship length overall (LOA). (**a**) 0–100 m LOA. (**b**) 100–200 m LOA. (**c**) Over 200 m LOA.

**Figure 10 sensors-22-08639-f010:**
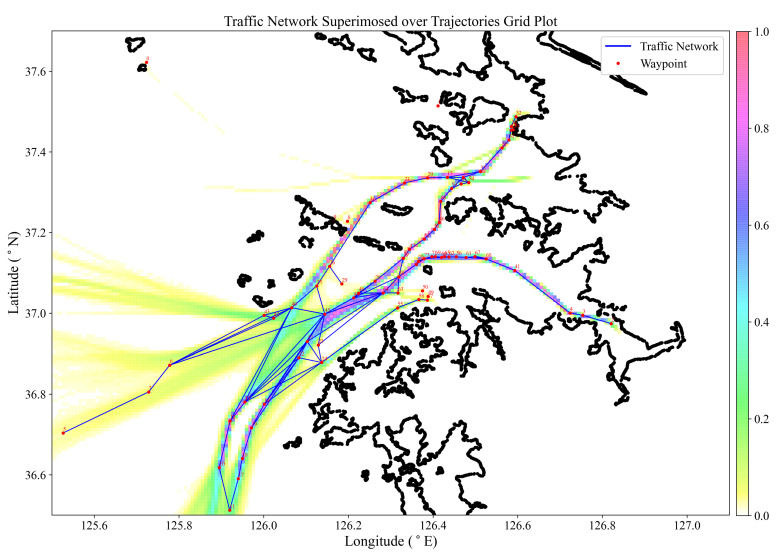
Maritime traffic network and waypoints superimposed on AIS trajectories grid plot for verification using SSPD metric.

**Table 1 sensors-22-08639-t001:** A summary of the approaches to ship route extraction by various authors.

Authors	Objective	Models Employed	Outcome
Filipiak et al. [8]	Automatic generation of maritime traffic network for ship routing and voyage planning	- CUSUM algorithm - GA algorithm - Graph algorithm	Traffic Network generated by comparative size of ship width, i.e., very large, large, medium, small and very small
Lee et al. [11]	Extraction of major nationalmaritime traffic route	- Marine Spatial Planning - Kernel Density Estimation	- Main route polygon - Outer branch route polygon - Inner branch route polygon
Zhang et al. [7]	Automatic maritime routegeneration algorithm	- DP algorithm - DBSCAN algorithm - ACA algorithm	Generation of optimal route
Forti et al. [3]	Extraction of maritime trafficpattern	- OU model - DBSCAN algorithm - Compact graph	Maritime traffic graph
Yan et al. [12]	Large area maritime traffic routes extraction	- Semantic trajectory extraction - OPTICS algorithm - Graph algorithm	Maritime traffic routes in open sea
Wen et. al. [1]	Automatic ship route designbetween two ports	- Course deviation for turning points - DBSCAN algorithm - ANN machine learning	Optimal path between two ports

**Table 2 sensors-22-08639-t002:** Transmission frequency of AIS messages in relation to dynamic information.

Navigation Status	Ship Speed	Transmission Period
At Anchor or Moored	<3 knots	3 min
	>3 knots	10 s
	0–14 knots	10 s
	0–14 knots and changing course	3.3 s
Cruising	14–23 knots	6 s
	14–23 knots and changing course	2 s
	>23 knots	2 s

**Table 3 sensors-22-08639-t003:** AIS data characteristics by ship type and length.

Number of trajectory observations	Number of ships	
49,686,178	3938	
**Ship Type**	**Number of ships**	**Proportion**
Passenger	24	0.6%
Cargo	2498	63.4%
Tanker	1416	36.0%
**Ship length**	**Number of ships**	**Proportion**
0–100 m	640	16.2%
100–200 m	2035	51.7%
Above 200 m	1263	32.1%

**Table 4 sensors-22-08639-t004:** Comparison of the sample routes from the maritime traffic network with the AIS trajectories.

HTSS-INCH			PYTK-SWST			INCH-HTSS			HTSS-DSN		
**Route**	**MMSI**	$D_{\mathrm{SSPD}}$	**Route**	**MMSI**	$D_{\mathrm{SSPD}}$	**Route**	**MMSI**	$D_{\mathrm{SSPD}}$	**Route**	**MMSI**	$D_{\mathrm{SSPD}}$
**Segment**	**Count**	**(m)**	**Segment**	**Count**	**(m)**	**Segment**	**Count**	**(m)**	**Segment**	**Count**	**(m)**
11–17	2769	327	12–31	1004	1143	32–34	1614	161	11–17	2769	327
17–18	3704	181	13–41	1398	1152	34–31	1838	186	17–18	3704	181
18–19	3884	146	14–41	1718	1170	31–30	1808	151	18–19	3884	146
19–14	3779	165	41–68	1841	1101	30–33	1760	147	19–14	3779	165
14–22	3220	285	68–67	2249	1182	33–37	1702	183	14–24	1158	238
22–27	3174	335	67–61	2238	1173	37–15	1051	184	24–55	1312	124
27–45	3188	358	61–56	2438	1180	15–20	3745	165	55–58	1192	153
45–44	3743	366	56–62	2474	1181	20–21	3822	139			
44–48	2836	330	62–65	2496	1175	21–16	3825	141			
48–28	2667	245	65–66	2519	1180	16–26	3796	178			
28–39	3053	173	66–69	2397	1184	26–25	3873	575			
39–47	3361	137	69–71	2262	1172	25–40	3430	423			
47–52	3319	131	71–73	2176	1198	40–13	1650	193			
52–51	3205	131	73–72	1531	1110	13–12	4063	197			
51–46	3179	131	72–70	1463	1110	12–10	4091	195			
46–60	3054	140	70–38	1792	1171						
60–59	2125	158	38–64	1707	1164						
59–37	1454	184	64–45	1556	1239						
37–33	1109	183	45–61	1606	1536						
33–30	1847	147	16–71	1791	1199						
30–31	1855	151	17–51	1630	1212						
31–34	1138	186									
34–32	1196	161									

## Data Availability

Not applicable.

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
