# Peer review of "A Quasi-Intelligent Maritime Route Extraction from AIS Data"

_sensors, 2022, doi:10.3390/s22228639_

Round 1

Reviewer 1 Report

This is an interesting paper and pretty well-written. It flows well and I think that someone without deep knowledge of the methods can still follow along.

COMMENTS:

PAGE 2, LINES 51-52: "clustering algorithms [3,7,11,12], Artificial Neural Network (ANN) [1]" should have an "and"

 Expand all acronyms on first use (e.g., MMSI)

 PAGE 4, LINES 111-113: Why are length and type of ship part of the set of "multidimensional temporal observations" about a ship's trajectory? They are static pieces of data. If they are important to trajectory, why not also add beam? In addition, would not SOG and ROT be important factors in trajectory?

PAGE 9: Reference is made to Figure 7(c) prior to reference to Figure 6, all followed by 7(a), 7(b), and 7(d). Is there any way to re-order the figures so that they better follow the flow of the paper? I know that this is a bit picky but it causes the reader to do a lot of jumping around.

Author Response

Comment 1: PAGE 2, LINES 51-52: “clustering algorithms [3,7,11,12], Artificial Neural Network (ANN) [1]” should have an “and”

Authors' response: Thank you for pointing this out. We have added the conjunction “and” before ‘Artificial Neural Network (ANN)’ on page 2, line 51 as suggested.

Comment 2: Expand all acronyms on first use (e.g., MMSI)

Authors’ response: Thank you for pointing this out. We have expanded the following acronyms on first use: SSPD as Symmetrized Segment-Path Distance on page 1, line 10; HDBSCAN as Hierarchical Density-Based Spatial Clustering of Applications with Noise on page 2, line 88; MMSI as Maritime Mobile Service Identity on page 3, line 101-102; IMO as International Maritime Organization on page 4, line 118 as suggested.

Comment 3: PAGE 4, LINES 111-113: Why are length and type of ship part of the set of "multidimensional temporal observations" about a ship's trajectory? They are static pieces of data. If they are important to trajectory, why not also add beam? In addition, would not SOG and ROT be important factors in trajectory?

Authors’ response: Indeed, the reviewer’s observation is correct. Static data is not associated with temporal features to be included in the trajectory. To account for this observation, we have amended the definition of the trajectory and adapted it to suit our study, as recommended in the two studies below. Adaptation herein refers to the selection of features that will be considered during maneuver detection in course variance analysis in section 2.2. Therefore, we have modified page 3, line 111 -117, Equation (1), and introduced two references.

  1. Laxhammar, R. Anomaly detection in trajectory data for surveillance applications. Doctoral dissertation, Örebro universitet, Örebro, 2011.
  2. Liu, B.; de Souza, E.N.; Matwin, S.; Sydow, M. Knowledge-based clustering of ship trajectories using density-based approach. In 2014 IEEE International Conference on Big Data (Big Data), 27 October 2014 (pp. 603-608). IEEE.

Comment 4: PAGE 9: Reference is made to Figure 7(c) prior to reference to Figure 6, all followed by 7(a), 7(b), and 7(d). Is there any way to re-order the figures so that they better follow the flow of the paper? I know that this is a bit picky but it causes the reader to do a lot of jumping around.

Authors’ response: Thank you for pointing this out. We have deleted the reference, figure 7(c), on page 9 line 259-260, and maintained the reference, figure 7(c), on page 10, line 301 to maintain flow of paper and arrangement of the figures.

Reviewer 2 Report

The authors proposed unsupervised machine learning algorithms for extracting shipping routes. The topic of this manuscript is very important in the maritime industry. The suggestions are listed as follow: (1) the literature review is not clear. I recommend that the authors add a table to summarize the previous studies. Even through, the authors proposed new methods for extracting shipping routes, in the experiment section, they need to compare the performance of their proposed method with previous studies.

Author Response

Comment 1: The literature review is not clear. I recommend that the authors add a table to summarize the previous studies.

Authors’ response: Thank you for this suggestion. We have added a tabulated summary of different approaches authors employ to construct maritime traffic networks as Table 1 on page 3.

Comment 2: Even through, the authors proposed new methods for extracting shipping routes, in the experiment section, they need to compare the performance of their proposed method with previous studies.

Authors’ response: Thank you for this suggestion. Studies on ship route extraction have used different distinct approaches which makes it quite a challenge to compare performance on a standardized scale. In spite of this challenge, we addressed the issue by comparative analysis of our study with previous studies as highlighted in line 365 – 378 on page 15.

Reviewer 3 Report

Point1: As shown in figure3, There are many obvious maneuver points that have not been identified. It means from the section 2.2 we have already get the wrong data, Not to mention the accuracy of the final results.

Point2: Section 2.5 not explain clearly, Please describe the how to get Maritime traffic networks

Point3:  In this paper, have not clear innovation points. Just using the “HDBSCAN Clustering”

Author Response

Comment 1: As shown in figure3, There are many obvious maneuver points that have not been identified. It means from the section 2.2 we have already get the wrong data, Not to mention the accuracy of the final results.

Authors’ response: Thank you for pointing this out. We used a large value of smoothing factor in course variance analysis and a moderate sensitivity threshold value that filtered out micro points to produce Figure 3. A large value of the smoothing factor filters out small course changes. Additionally, a high value of sensitivity threshold risk skipping significant maneuvers. Therefore, to demonstrate how robust our proposed approach is, with the same data set, we have made changes to all experiment figures and design approaches as follows:

  1. We selected a lower value of the smoothing factor and a low sensitivity threshold to discover maneuver points as shown in Figure 3, page 6.
  2. We selected the lowest sensitivity threshold of 0.05, compared to 0.15 in original draft, for subsequent clustering and traffic network construction. As a result, the selection of HDBSCAN min cluster size was adjusted to 119 as shown in Figure 7(a) and Figure (b) on page 11. Moreover, a total of 74 clusters were discovered as highlighted in Figure 7 (d) from maneuver plot in Figure 7 (c) on page 11, and subsequently, results in Figure 8(a), (b), (c), (d), and Figure 9(a), (b), and (c) extracted as displayed on page 12 and page 13 respectively. Selection of a low sensitivity threshold resulted in additional waypoints, and subsequently increased the number of networks as evidenced in figure 10 page 14, and evaluation in table 4 page 15, as compared to original study, thereby improving the route extraction accuracy.

Comment 2: Section 2.5 not explain clearly, Please describe the how to get Maritime traffic networks

Authors’ response: In section 2.5 we strived to paraphrase the network construction procedure from python code in a logical manner. Therefore, we have added a pseudocode as appendix A on page 17 to supplement description of maritime traffic network construction.

Comment 3: In this paper, have not clear innovation points. Just using the “HDBSCAN Clustering”

Authors’ response: Thank you for pointing this out. The innovation in our approach is on these three fonts;

  1. Selection of HDBSCAN min cluster size using clustering performance metric.
  2. Construction of maritime traffic network to display prominent routes quantitatively by ship characteristics.
  3. Combination of HDBSCAN clustering and graph theory for maritime traffic network construction.

We have amended the conclusion part, line 384 – 389 on page 16, to point out the innovations in this paper.

Round 2

Reviewer 2 Report

All of comments have been well responded by the authors.

Author Response

We have moderated the article by checking for grammatical errors, flow process in the proposed approach explanations, and added more details in the discussion to improve the article.

Reviewer 3 Report

1.     The research area is containing the Ship routing system, thus the maritime route network was similar with the Ship routing system

2.     As for the Fig10, please explain the waypoint0, waypoint1, waypoint8 and waypoint9

3.     Please explain the difference between the extraction route with the AIS data statistics results

4.     Please explain why the range about the Point70-72-73-69-66-65-62-56-61-67-68 is so dense, but the range about the Point40-25-26-16-21-20 is so sparse

Author Response

Comment 1: As for the Fig10, please explain the waypoint0, waypoint1, waypoint8 and waypoint9.

Authors’ response: Thank you for this observation. The isolated waypoints 0, 1, 8, and 9, are minor local landing sites and routes with low traffic to construct a traffic network in relation to the area network, but significant maneuver points for waypoint identification. We have added a detailed explanation in lines 370 – 372, page 15.

Comment 2: Please explain the difference between the extraction route with the AIS data statistics results

Authors’ response: In section 4, we have introduced a distance-based metric, SSPD, to measure the deviation of actual ship behavior (AIS data grid plot) to our constructed traffic network. The measurement standard is that the smaller the SSPD values the better the performance. The details are introduced in lines 343 – 354, pages 13 and 14, and specific explanations are provided in lines 360 – 370 on page 15.

Comment 3: Please explain why the range about the Point70-72-73-69-66-65-62-56-61-67-68 is so dense, but the range about the Point40-25-26-16-21-20 is so sparse

Authors’ response: Thank you for pointing this out. The area at point 70-72-73-69-66-65-62-56-61-67-68 is part of a two-way traffic route to and from Pyeongtaek port. Therefore, the dense cluster waypoints are from the course keeping maneuvers from both inbound and outbound ships at the Masan-sudo channel bends. Point 40-25-26-16-21-20 is a one-way outbound traffic from Incheon which results in sparsely spaced waypoints.
We have added the observation into the manuscript in order to improve the quality of the results and discussion in lines 364 – 367, page 15.